



**Effects of temperature on the composition and diversity of bacterial**
**communities in bamboo soils at different elevations**
Yu-Te Lin[1], Zhongjun Jia[2], Dongmei Wang[2], Chih-Yu Chiu[1,*]
[1] Biodiversity Research Center, Academia Sinica, Taipei 11529, Taiwan
[2] State Key Laboratory of Soil and Sustainable Agriculture, Institute of Soil Science,
Chinese Academy of Science, Nanjing, China
* Corresponding author: Biodiversity Research Center, Academia Sinica, Taipei,
Taiwan. Tel: +886-2-2787-1068; E-mail: bohiu@sinica.edu.tw.





**ABSTRACT**
Bamboo is an important resource distributed in mountain areas in Asia. Little is
known about the impact of temperature changes on bamboo soil bacterial
communities. In this study, responses of bacterial communities collected at 600,
1,200, and 1,800 m to different incubation temperatures (15°C, 20°C, and 35°C)
were examined using barcoded pyrosequencing and soil analyses. Soil respiration
was greater at higher elevation and incubation temperature. The bacterial diversity in
samples incubated at 35°C decreased after 112 days of incubation. Before incubation,
*Acidobacteria* and *Proteobacteria* were the most abundant phyla in all communities.
The relative abundance of *Acidobacteria* generally decreased after 112 days of
incubation at the three temperatures. α-*Proteobacteria* showed a similar trend, while
the abundance of γ-*Proteobacteria* increased after incubation, except those from
1,800 m incubated at 35°C. Non-metric multi-dimensional scaling analysis revealed
structural variability under different incubation times and temperatures. Principal
component analysis indicated that bacterial structure incubated at 35°C correlated
with temperature and soil respiration, while structures in samples incubated at 15°C
and 20°C correlated with time. These results suggest that a temperature rise could
result in increasing soil respiration and soil soluble carbon and nitrogen
consumption, influencing the bacterial diversity and structure at different elevations.
*Keywords:* temperature, bamboo, soil



# 1. Introduction

Temperature is known as one of the most important factors influencing soil organic matter decomposition and microbial communities. For example, temperature significantly affects the soil microbial phospholipid fatty acid composition associated with straw decomposition at the early stage (Zhou et al., 2016). Bacterial abundance increases in conditions of elevated temperature and $CO_2$ concentration (Castro et al., 2010). The complex responses of bacterial composition and diversity of bamboo soils across altitudinal gradients have been suggested to result from interactions with multiple factors, including temperature (Lin et al., 2015).

In Taiwan, moso bamboo (*Phyllostachys pubescens*) is an important versatile forest resource that is widely used for food and construction and as a furniture material. It distribute from low mountain region to high mountain at about 1,800 m a.s.l. Management practices for increasing bamboo production, including regular removal of understory vegetation, tillage, and fertilizer application, could increase the soil $CO_2$ efflux (Liu et al., 2011) and water-soluble organic N concentration (Wu et al., 2010). Because of the decrease in bamboo prices, many bamboo plantations are in an unmanaged condition. Under the unmanaged conditions, the aboveground biomass of old bamboo has been sharply increasing (Chen et al., 2016). Considering the effects of environmental factors, it is worth elucidating the changes in bamboo soil bacterial communities, under managed and unmanaged conditions.

Our previous study revealed that bamboo invasion could increase bacterial diversity and alter the bacterial structure of adjacent cedar forest soils (Lin et al., 2014). Soil bacterial diversity in bamboo plantations showed a hump-backed trend, and community structure formed different clusters along with elevations (Lin et al., 2015). Our parallel study showed that bamboo increased humification of soil organic





matter (SOM) (Wang et al., 2016b), In addition, changes in the SOM pool and the
rate of humification with elevation were primarily affected by changes in climatic
conditions along the elevation gradient in the bamboo plantations (Wang et al.,
2016a). However, it is not known whether bamboo soil bacterial groups respond to
the temperature changes.

6        Bacteria could have distinct ecological classification that different phylogenetic

groups could represent different functional groups, and their relative abundance
affected by C availability. For example, as copiotroph, the relative abundance of
*Proteobacteria* is more abundant under C rich environment. In contrast, oilgotrophic
groups (e.g., *Acidobactreria*) could maintain viability under stressful environmental
conditions (Fierer et al., 2007). However, little is known about how these two
distinct groups respond to the environment changes caused by temperature. Here, we
hypothesized that the temperature changes would alter the structure and diversity of
soil bacterial communities at different elevations, and that bacterial taxa, including
copiotrophic and oligotrophic groups, would reveal distinct responses to different
nutrient availability cause by temperature changes. To test these hypotheses, soil
communities sampled at bamboo plantations at three elevations were incubated at
different temperatures and investigated by using the barcoded pyrosequencing
technique. The objectives of this study were to elucidate (1) changes in soil organic
carbon, nitrogen, and respiration at elevation gradients and at different incubation
temperatures, (2) differences in bacterial structure and diversity under different
incubation temperatures and periods, and (3) responses of different phylogenetic
groups to the temperature changes.
**2. Methods**

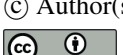



*2.1. Site description and soil sampling*
This study was conducted in Mt. Da-an, a subtropical montane area in Nantou
County, central Taiwan (23°42' N, 120°41' E). The soil samples were collected from
moso bamboo plantations at 600, 1,200, and 1,800 m a.s.l. along a county road. The
three sampling sites were all dominated by moso bamboo with few understory plants.
Based on the record from weather stations and temperature-elevation correlation, the
annual mean air temperature was estimated as 20.3°C at 600 m, 17.2°C at 1,200 m,
and 14.1°C at 1,800 m with a decrease of 0.52°C per 100 m elevation gain (Wang et
al., 2016a). At each elevation, three 25 × 25 m plots were established along transect
lines in March 2015. Within each plot, three subsamples were collected with a soil
auger 8 cm in diameter and 10 cm deep and pooled. Visible detritus, such as roots
and litter, were manually removed prior to passing the soil through a 2-mm sieve.
Soil samples collected at each elevation were combined and homogenized for
further incubation and analysis. The sieved soils were stored at 4°C before
incubation experiments.
*2.2. Incubation experiment and soil analysis*
The three replicates (25 g for each replicate) from each elevation were incubated
at 15°C, 20°C, and 35°C for 112 days. Temperature 15°C, and 20°C were selected
based on the mean annual temperature, while 35°C was selected to simulate the
summer condition. During the entire incubation period, the soil moisture was
maintained at 60% of the water-holding capacity. Soil samples of different
incubation times were taken from the same container. Soil respiration ($CO_2$-C) was
measured as described (Huang et al., 2014). Soluble organic carbon (SOC) and


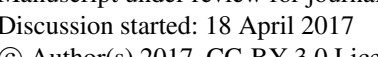


nitrogen (SON) were extracted from the soil samples at different incubation times
with 2 M KCl, and measured with the Fisons NA1500 elemental analyser
(ThermoQuest Italia, Milan, Italy) as described (Huang et al., 2014).
*2.3. Barcoded pyrosequencing of the 16S rRNA genes*
Soil community DNA was extracted using the PowerSoil® Soil DNA Isolation kit
(MoBio Industries, Carlsbad, CA, USA) in accordance with the manufacturer's
instructions. The V1 to V2 regions of the bacterial 16S rRNA gene were amplified
using 27F and 338R primers (Lane, 1991). Polymerase chain reactions (PCR) were
performed as described previously (Lin et al., 2015). Secondary PCR (using 3 cycles
instead of 20) was carried out to barcode the DNA in each sample. The unique and
error-correcting bar codes facilitated sorting of sequences from a single
pyrosequencing run (Hamady et al., 2008). The barcoded PCR products were
purified on a column filter using a PCR cleanup system (Viogene Biotek Corp., New
Taipei City, Taiwan). The qualities and concentrations of the purified barcoded PCR
products were determined using a NanoDrop Spectrophotometer (Thermo Fisher
Scientific, Waltham, MA, USA). Amplicon pyrosequencing was performed by
Mission Biotech (Taipei, Taiwan) using the 454/Roche GS-FLX Titanium
Instrument (Roche, Branchburg, NJ, USA). All sequences have been submitted to
the Short Read Archives under accession number SRS1923345.
*2.4. Sequence analyses*
The pyrosequences were processed through the RDP pyrosequencing pipeline
(http://pyro.cme.msu.edu; RDP Release 11.5; release date: 2016.09.30). The





sequences were assigned to the samples by recognition of the bar code from a tag file,
followed by trimming of bar codes, primers, and linkers. The pyrosequences were
filtered, and sequences that did not contain Ns, were more than 200 bp in length, and
possessed quality scores >25 were selected for further analyses. Taxonomic
information was analysed using the naïve Bayesian rRNA classifier in RDP (Wang et
al., 2007). The Shannon diversity index was calculated based on Complete Linkage
Clustering data for operational taxonomic units (OTUs), with an evolutionary distance
of 0.03. The distribution of shared OTUs among the communities was obtained using
the Mothur program (Schloss et al., 2009). The abundance-based Jaccard similarities
among communities (β-diversity) at an evolutionary distance of 0.03 was computed
with Mothur program. Pairwise similarity values were converted to distances and used
to construct dendrogram. Non-metric multi-dimensional scaling (NMDS) based on the
distribution of shared OTUs was plotted by using the PRIMER V6 software (Clarke &
Gorley, 2006). The Mantel tests as implemented in PRIMER V6 software was used to
analyse the relationships between bacterial communities, phylogenetic groups and soil
properties. Principal component analysis (PCA) to determine the relationship between
bacterial community and soil properties was carried out using R v.3.2.1.
**3. Results**
*3.1. Soil respiration, SOC and SON*
The results of soil respiration $CO_2$-C in samples taken from three elevations and
incubated at different temperatures are shown in Fig. 1. Under the same temperature,
the soil samples collected at higher elevation, especially those from 1,800 m, had a
significantly higher soil respiration rate than those obtained at lower elevation. The



soil respiration rate was also increased with temperature within each elevation. At
35°C, the soil respiration rate decreased significantly with incubation time. At 15°C
and 20°C, the respiration rates of some soil samples slightly increased in the early
incubation period (d28) (Fig. 1). Because the respiration rate after d72 was similar,
we only applied the rate up to d72 for further analysis.
At the beginning of incubation at day 0 (d0), the SOC and SON contents of the
soils increased significantly with elevation (Fig. 2). Compared to d0, the
concentration of SOC in the high-elevation soils (1,800 m) decreased, while those at
600 and 1,200 m increased after 112 days (d112) of incubation at three temperatures.
Incubation at higher temperature (35°C) resulted higher SOC content than that at
lower temperature (15°C and 20°C). In most samples, SON content increased over
the first 28 days (d28) of incubation, but decreased at d112.
*3.2. Community diversity at different temperatures*
The soil bacterial diversity at three elevations at different incubation temperatures
was determined based on the OTUs formed at an evolutionary distance ≤ 0.03.
Based on the Shannon diversity index, the bacterial diversity of soils incubated at
35°C decreased after long incubation (d112). Under incubation at 15°C or 20°C, the
bacterial diversity slightly increased at d7 and d28, and decreased at d112 (Fig. 3).
Analysis of the β-diversity revealed that though incubated with different temperature,
the communities at the same elevation formed a cluster different from those at other
elevation (Supplementary Fig. 1).
*3.3. Community composition at different incubation temperatures*



Before incubation, *Acidobacteria* and *Proteobacteria* were the two most
abundant phyla in soils from all three elevations, together representing more than
60% of the soil bacterial communities (Fig. 4a). Within the *Proteobacteria*,
α-*Proteobacteria* were predominant (Fig. 4b). At 1,800 m, *Bacteroidetes* accounted
for 8% of the community, while they comprised only 2–4% of the communities at
the two other elevations. The relative abundance of *Actinobacteria* was 4–6%, and
the other phylogenetic groups represented less than 3% of the communities.
Bacterial groups of the soil communities showed different responses to the
incubation temperature. The relative abundance of *Acidobacteria* at 600 and 1,200 m
gradually decreased over the entire incubation period at all temperatures (Fig. 5a-5f).
At 1,800 m, it increased during the first seven days of incubation at 35°C, and
decreased thereafter at all temperatures (Fig. 5i). The relative abundance of
α-*Proteobacteria* showed similar trends; it gradually decreased at 600 and 1,200 m
over the entire incubation period at different temperatures, except at d7 at 600 m,
20°C, and at d7 at 1,200 m, 35°C (Fig. 5a-5f). At 1,800 m, the changes in abundance
were different. α-*Proteobacteria* were elevated at d7 and d112, but were lower at
d28 of incubation at 15°C and 20°C. Their abundance decreased over time under
incubation at 35°C (Fig. 5g-5i). With regard to γ-*Proteobacteria*, their relative
abundance mostly increased over incubation, except in soils sampled at 1,800 m
under incubation at 35°C, in which it was increased at d7, but decreased at d28 and
d112 (Fig. 3c). The relative abundance of *Chloroflexi* also increased over incubation,
except that at 600m incubated at 15 °C, d112. Some other phyla demonstrated
inconsistent   changes   under   increased   temperature.   The   abundances   of
*Actinobacteria* at 1,200 m and 1,800 m increased at higher temperature (Fig. 5d-5i),
while it decreased in samples taken at 600 m (Fig. 5a-5c). Likewise, *Bacteroidetes*



showed inconsistent changes after different incubation times and temperatures (Fig.
5a-5i).
NMDS analysis based on the distribution of shared OTUs also revealed the
variability in bacterial structure under different incubation times and temperatures
(Fig. 6). The bacterial community at 1,800 m formed a different cluster from those at
600 and 1,200 m. Incubation at higher temperature (35°C) led to a bacterial structure
different from those at 15°C and 20°C. Incubation time also changed the bacterial
structure. The bacterial structure at long incubation time (d112) was different from
those at d7 and d28.
PCA analysis revealed the correlation between bacterial structure and
environmental factors. When incubated at 35°C, bacterial structure correlated with
temperature and soil respiration $CO_2$-C, while at 15°C and 20°C, bacterial structure
correlated with incubation time (Fig. 7).
**4. Discussion**
The present study revealed that the SOC content was higher at high incubation
temperature and decreased at higher elevation after long incubation. The soil
respiration $CO_2$-C rate was greater at higher elevation. Similarly, a previous study in
tundra soils using different incubation temperatures reported higher respiration rate
at high temperatures (Stark et al., 2015). Incubation at increasing temperatures
enhanced the soil microbial activity and led to an increase in soil respiration in forest
mesocosms (Lin et al., 2001). In our study, the respiration rate decreased after long
incubation. This could be due to the exhaustion of labile compounds after microbial
decomposition (Zhou et al., 2016). The decrease in bacterial diversity at high
elevation and high incubation temperature could also be the result of nutrient




exhaustion after long incubation. In addition, the correlation between soil respiration
and bacterial structure in the soil samples under incubation at 35°C suggests the
adaption and high activity of bacterial communities at higher temperature.

4        The bacterial community structure varied over different incubation periods and

temperatures. The communities at the three elevations formed different clusters as
compared to the results of our previous study (Lin et al., 2015). Soils at different
elevation have distinct soil SOC and SON contents, which could result in different
forces to alter bacterial communities. Incubation temperature had an effect on
community structure. Warming in the experimental field in a previous study in the
Arctic environment caused a significant increase in the abundance of fungi and
bacteria (Yergeau *et al.*, 2012). The quantity of SOC and $CO_2$ flux has been shown
to increase under warming condition (Zhang et al., 2005; Zhou et al., 2011).
Increasing temperature increased relative bacterial growth in arable soils from
southern Sweden (Bárcenas-Moreno et al., 2009), and particularly, the abundance of
genes involved labile carbon degradation in a tall-grass prairie ecosystem in Central
Oklahoma, USA, and led to C loss. In the present study, the shifts in bacterial
communities at three elevations could reflect differences in nutrient availability,
including SOC and SON, and bacterial activity under different incubation
temperatures and at distinct time points during incubation.

20       Bacterial community structure under incubation at 35°C was affected by

temperature, while under incubation at 15°C and 20°C, it correlated with incubation
time (Fig. 5). Warming has been shown to change the bacterial structure of alpine
meadow soils (Xiong et al., 2014) and to cause thermal adaption in functional shift
of microbial communities (Rousk et al., 2012). Recent studies have observed
changes in temperature sensitivity of microbial communities along incubation time.
Shifts in microbial communities in response to warming occur after a few years





(Yergeau et al., 2012) or even only a few months (Xiong et al., 2014). However,
some studies revealed no significant community changes due to warming across
time (Allison et al., 2010; Zhou et al., 2011). The present work revealed community
structure differences after incubation for only about four months, suggesting that the
bacterial communities in bamboo soils at elevation are highly sensitive to
temperature changes, even though they faced a relative short-time warming
condition.
The responses of phylogenetic abundances to temperature differed. As for
*Acidobacteria*, the abundance generally decreased with increasing temperature. This
is in accordance with previous studies showing decreases in the relative abundance
of *Acidobacteria* in warming soils (Xiong et al., 2014; Yergeau et al., 2012).
*Acidobacteria* are known as slow-growing (oligotrophic) bacteria that prefer low
nutrient availability (Fierer et al., 2007). Warming conditions in the soil could
increase substrate availability and might favour fast-growing (copiotrophic)
microorganisms. Thus, the decreases in the abundance of *Acidobacteria* could
reflect their interactions with copiotrophic species. This phylum may be an indicator
of climate warming in soil ecosystems (Xiong et al., 2014).
Under increased temperature, some phyla in our study responded differently
from previous studies. Increasing α-*Proteobacteria* abundance has been observed in
short warming conditions (Xiong et al., 2014) and in a range of Antarctic
environments (Yergeau et al., 2012). α-*Proteobacteria* are mostly fast-growing
(copiotrophic) bacteria, and known to be positively correlated with soil available C
pools (Nemergut et al., 2010). The decreases in the abundance of α-*Proteobacteria*
in the present study could reflect the decrease in SOC content, which was exhausted
by soil respiration $CO_2$-C after incubation. Increased in the abundance of
γ-*Proteobacteria* after incubation in our study also differed from that in the soil





community subjected elevated soil temperature. The γ-*Proteobacteria* showed a
lower relative abundance under elevated temperature treatment compared to ambient
control (Ren et al., 2015). In addition, *Actinobacteria* and *Bacteroidetes* showed
variable responses at different temperatures. These phyla also prefer nutrient-rich
environments (Nemergut et al., 2010). Differences in vegetation and litter quality
among the study sites might explain this variation. The results of previous study
suggest the relationship of elevation and temperature to the decomposition of
recalcitrant C (Wang et al., 2016a). After decomposition of labile C, the availability
of recalcitrant C could also serve an important factor to affect the community.
Moreover, based on the literature survey by Ho et al. (2017), the consistency in the
oligotrophic and copiotrophic phyla of bacterial communities is little. The
microorganisms could process a variety of metabolic characteristics; adjust between
high and low substrate use efficiency, to adapt environmental changes. Therefore,
shifts in the relative abundances of bacterial taxa may not necessarily indicate their
life strategies in oilgotroph or copiotroph. It would just reveal the response of
community to the local factors (Ho et al., 2017). Further study, including more
comprehensive temperature gradients and more detailed time course analysis, will
be necessary to elucidate the exact influences of temperature to soil communities. In
addition, an interesting pattern was shown at some bacterial groups. The
*Acidobacteria* and α-*Proteobacteria*, comprised more than 10% of the communities
before incubation, revealed decreasing response in relative abundance after
incubation. The groups with lower abundance of communities before incubation,
especially γ-*Proteobacteria*, responded in increasing trend after incubation. This
patter was similar to that shown in communities of a rice paddy and desert soils
(Wang et al., 2012; Ren et al., 2015). The numerically dominant bacterial
phyla/classes were reduced, while original rare groups increased in relative

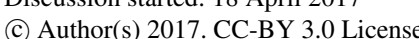



abundance after exposed to environmental changes. These results suggest the shifts
of the bacterial populations faced to the environmental changes may follow a
predictable pattern. The dominant bacterial groups will become less abundant or
even rare taxa, while initial less/rare abundant groups will become dominant after a
period of incubation time (Ren et al., 2015).
In conclusion, our results revealed that an increase in temperature could result in
increased soil respiration $CO_2$-C and consumption of SOC and SON contents, which
directly or indirectly influence the bacterial diversity and structure of bamboo soils
at different elevations. In addition, the different responses of bacterial groups to the
temperature changes suggest the adaptation of soil communities to global
warming-related climatic changes. This study highlights the need for further
research on the physiologic and ecologic roles of soil bacterial members, such as
*Acidobacteria*, α- and γ-*Proteobacteria*, in climatic change in forest ecosystems.





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

17       barcoded primers for pyrosequencing hundreds of samples in multiplex. Nat
18       Methods 5, 235-237.
Ho, A., Lonardo, D.P.D., Bodelier, P.L.E. 2017. Revisiting life strategy concepts in
20       environmental microbial ecology. FEMS Microbiol Ecol 93, fix006.
Huang, C.Y., Jien, S.H., Chen, T.H., Tian, G., Chiu, C.Y. 2014. Soluble organic C
22       and N and their relationships with soil organic C and N and microbial
23       characteristics in moso bamboo (*Phyllostachys edulis*) plantations along an
24       elevation gradient in Central Taiwan. J Soils Sediments 14, 1061-1070.
Lane, D. J. 1991. 16S/23S rRNA sequencing. In: Stackbrandt E, Goodfellow M (eds)
Nucleic acid techniques in bacterial systematics. Wiley, New York., pp 115-175.
Lin, G., Rygiewicz, P.T., Ehleringer, J.R., Johnson, M.G., Tingey, D.T. 2001.
Time-dependent responses of soil $CO_2$ efflux components to elevated
atmospheric $[CO_2]$ and temperature in experimental forest mesocosms. Plant
Soil 229, 259-270.
Lin, Y.T., Tang, S.L., Pai, C.W., Whitman, W.B., Coleman, D.C., Chiu, C.Y. 2014.
Changes in the soil bacterial communities in a cedar plantation invaded by moso
bamboo. Microb Ecol 67, 421-429.
Lin, Y.T., Whitman, W.B., Coleman, D.C., Shi, S.Y., Tang, S.L., Chiu, C.Y. 2015.
Changes of soil bacterial communities in bamboo plantations at different
elevations. FEMS Microbiol Ecol 91, fiv033.
Nemergut, D.R., Cleveland, C.C., Wieder, W.R., Washenberger, C.L.,& Townsend,
38       A.R. 2010. Plot-scale manipulations of organic matter inputs to soils correlate




with shifts in microbial community composition in a lowland tropical rain forest.
Soil Biol Biochem 42, 2153-2160.
Ren, G., Zhu, C., Alam M.S., Tokida, T., & Sakai, H., Nakamura H., Usui Y., Zhu J.,
Hasegawa, T., & Jia, Z. 2015. Response of soil, leaf endosphere and phyllosphere
bacterial communities to elevated $CO_2$ and soil temperature in a rice paddy. Plant
Soil 392, 27-44.
Rousk, J., Frey, S.D., Bååth, E. 2012. Temperature adaptation of bacterial
communities in experimentally warmed forest soils. Glob Change Biol 18,
9    3252-3258.
Schloss, P.D., Westcott, S. L., Ryabin, T. & other authors 2009. Introducing mothur:
Open-Source, platform-Independent, community-supported software for
describing and comparing microbial communities. Appl Environ Microbiol 75,
13    7537-7541.
Stark, S., Männistöb, M.K., Ganzert, L., Tiirola, M., Häggblom, M.M. 2015.
Grazing intensity in subarctic tundra affects the temperature adaptation of soil
microbial communities. Soil Biol Biochem 84, 147-157.
Wang, B.Z., Zhang, C.X., Liu, J.L., Zeng, X.W., Li, F.R., Wu, Y.C., Lin, X.G.,
Xiong, Z.Q., Xu, J., Jia, Z.J. 2012. Microbial community changes along a
land-use gradient of desert soil origin. Pedosphere 22, 593-603.
Wang, H.C., Chou, C.Y., Chiou, C.R., Tian, G., Chiu, C.Y. 2016a. Humic acid
composition and characteristics of soil organic matter in relation to the elevation
gradient of moso bamboo plantations. PLoS ONE 11, e0162193.
Wang, H.C., Tian, G., Chiu, C.Y. 2016b. Invasion of moso bamboo into a Japanese
cedar plantation affects the chemical composition and humification of soil
organic matter. Sci Rep 6, 32211.
Wang, Q., Garrity, G.M., Tiedje, J.M., Cole, J.R. 2007. Naïve Bayesian Classifier for
rapid assignment of rRNA sequences into the new bacterial taxonomy. Appl
Environ Microbiol 73, 5261-5267.
Xiong, J., Sun, H., Peng, F., Zhang, H., Xue, X., Gibbons, S.M., Gilbert, J. A.,Chu,
H. 2014. Characterizing changes in soil bacterial community structure in
response to short-term warming. FEMS Microbiol Ecol 89, 281-292.
Yergeau, E., Bokhorst, S., Kang, S., Zhou, J., Greer, C.W., Aerts, R.,& Kowalchuk,
G.A. 2012. Shifts in soil microorganisms in response to warming are consistent
across a range of Antarctic environments. ISME J 6, 692-702.
Zhang, W., Parker, K.M., Luo, Y., Wan, S., Wallace, L.L., Hu, S. 2005. Soil
microbial responses to experimental warming and clipping in a tallgrass prairie.
Glob Change Biol 11, 266-277.
Zhou, G., Zhang, J., Chen, L., Zhang, C., Yu, Z. 2016. Temperature and straw



quality regulate the microbial phospholipid fatty acid composition associated
with straw decomposition. Pedosphere 26, 386-398.
Zhou, J., Xue, K., Xie, J., other authors 2011. Microbial mediation of carbon-cycle
feedbacks to climate warming. Nat Clim Change 2, 106-110.
**Acknowledgements**
The authors thank the Ministry of Science and Technology of Taiwan, Republic of
China, for financially supporting this research under contract number MOST
105-2621-B-001-007. The authors are also grateful to Ms. Yu-Shiuan Huang from
Biodiversity Research Center, Academia Sinica, Taipei, Taiwan for molecular
analyses.
**Author Contributions**
YTL performed statistical analyses, ZJ built statistical models. CYC interpreted
ecological rationale. ZJ and CYC formulated the study hypothesis and developed the
methodology. YTL wrote, and ZJ and CYC edited the manuscript. All authors read
and approved the final manuscript.
**Competing financial interests**
The authors declare that they have no competing interests.



**Figure legends**
**Fig. 1.** Respiration $CO_2$-C rate in soils sampled at three elevations and incubated at
(a) 15 °C, (b) 20 °C and (c) 35 °C. Error bars represent standard deviation.
**Fig. 2.** Concentration of (a-c) soluble organic carbon (SOC) and (d-f) nitrogen (SON)
in bamboo soils sampled at three elevations and incubated at (a, d) 15 15 °C, (b, e)
20 °C and (c, f) 35 °C. Error bars represent standard deviation.
**Fig. 3.** Changes in bacterial diversity of soil community at 600 m, 1,200 m, and
1,800 m incubated at different temperatures.
**Fig. 4.** Relative abundances of (a) all phylogenetic groups and (b) all phylogenetic
groups except *Acidobacteria* in the bamboo soil bacterial communities at different
elevations.
**Fig. 5.** Changes in relative abundance of phylogenetic groups of bamboo soil
bacterial communities at (a) 600 m, 15 °C, (b) 600 m, 20 °C, (c) 600 m, 35°C, (d)
1,200 m, 15 °C, (e) 1,200 m, 20 °C, (f) 1,200 m, 35 °C, (g) 1,800 m, 15 °C, (h)
1,800 m, 20 °C and (i) 1,800 m, 35 °C. Abbreviation: Acid: *Acidobacteria*; Actino:
*Actinobacteria*; Bac: *Bacteroidetes*; Chloro: *Chloroflexi*; Firm: *Firmicutes*; Gem:
*Gemmatimonadetes*; Nitro: *Nitrospirae*; α, β, γ, δ: α-, β-, γ- and δ-*Proteobacteria*.
**Fig. 6.** NMDS analysis of bamboo soil bacterial communities sampled at three
elevations and incubated at different temperatures. Circles, triangles, and diamonds
represent communities at 600 m, 1,200 m and 1,800 m elevation, respectively. The



analysis was based on the distribution of OTUs formed at an evolutionary distance
of 0.03.
**Fig. 7.** PCA analysis of bamboo soil bacterial communities and environmental
properties. Symbols are the same as in Fig. 4.





2    **Fig. 1.**

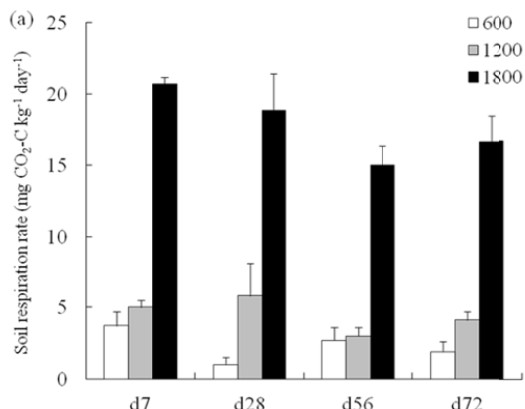

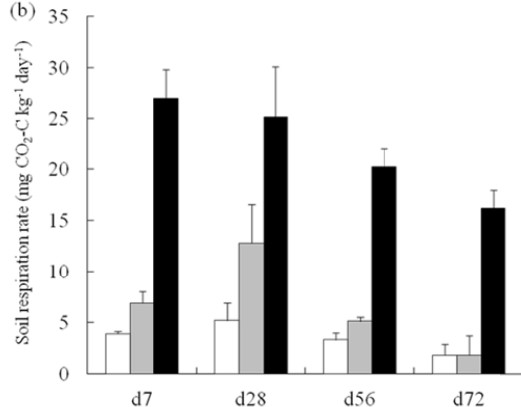

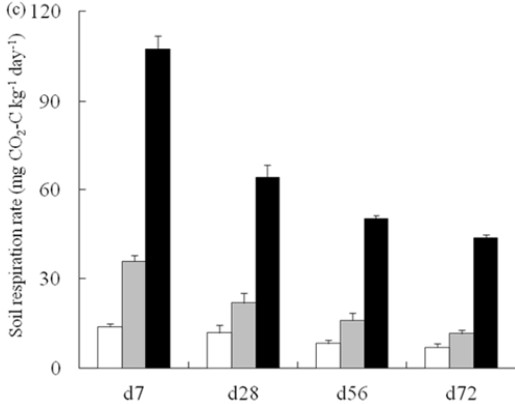



**Fig. 2.**

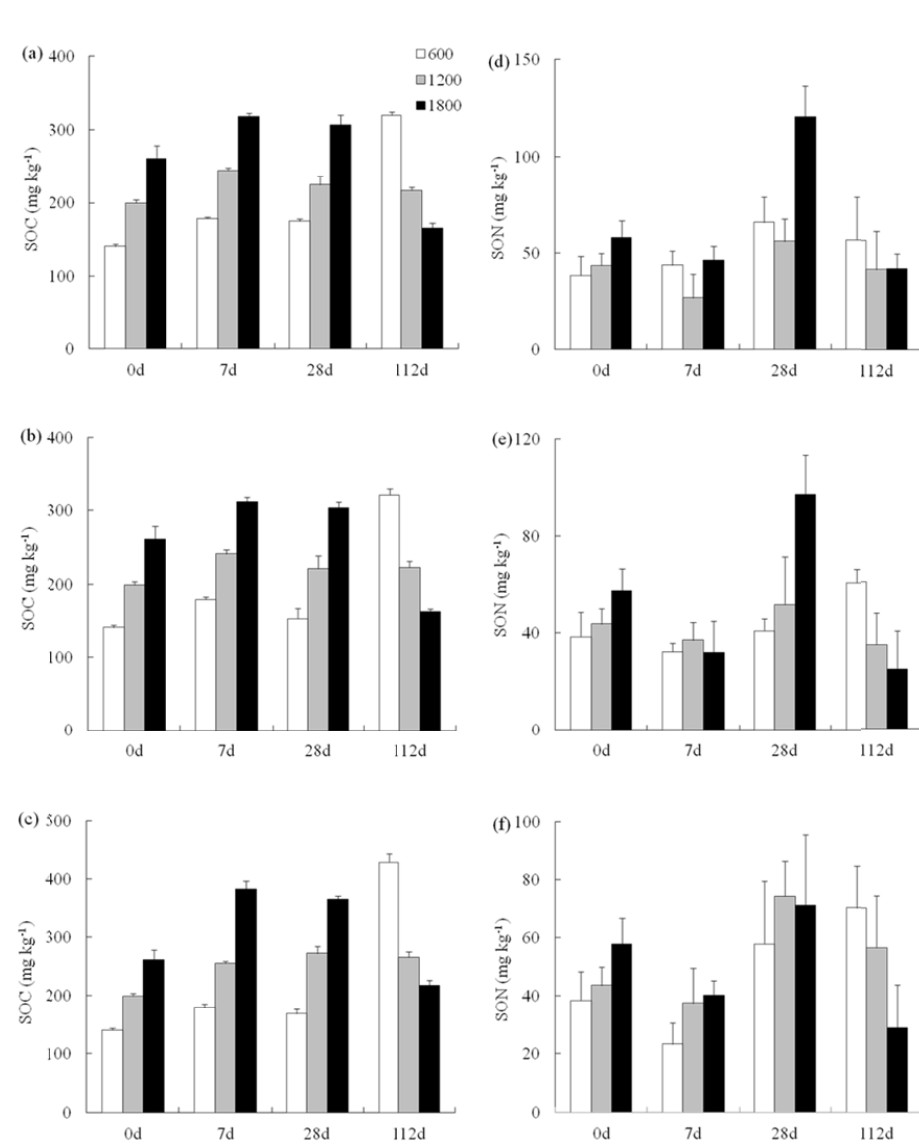



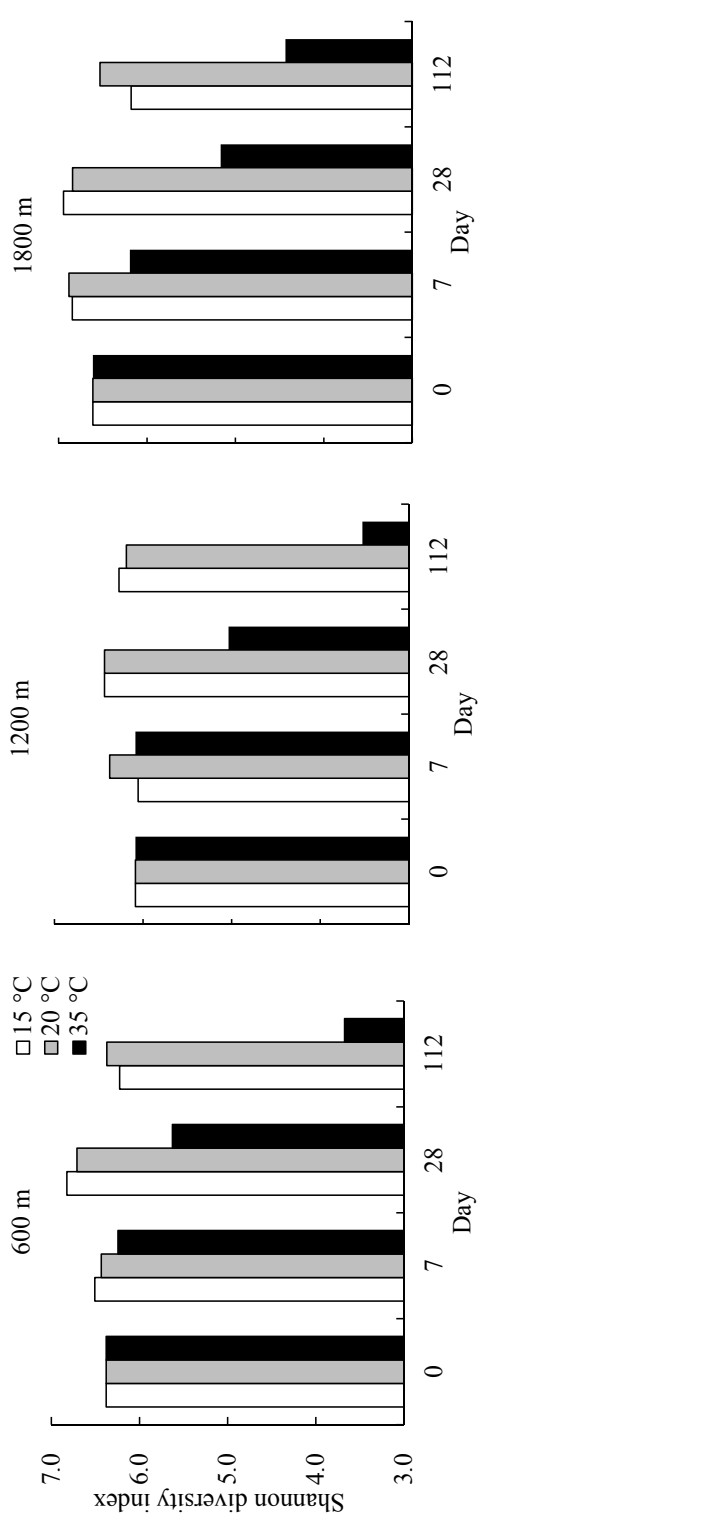

**Fig. 3.**




1    **Fig. 4**

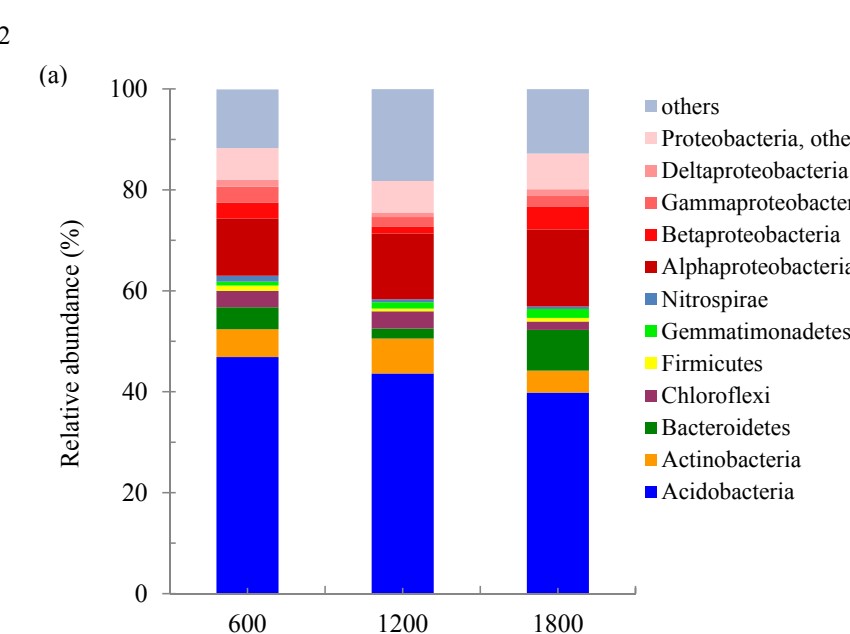

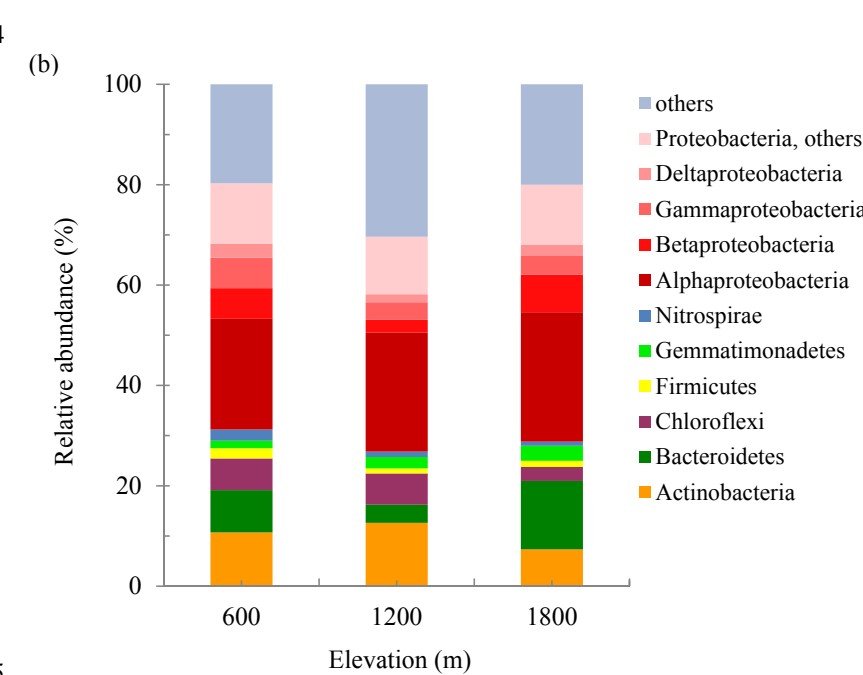



2 **Fig. 5.**

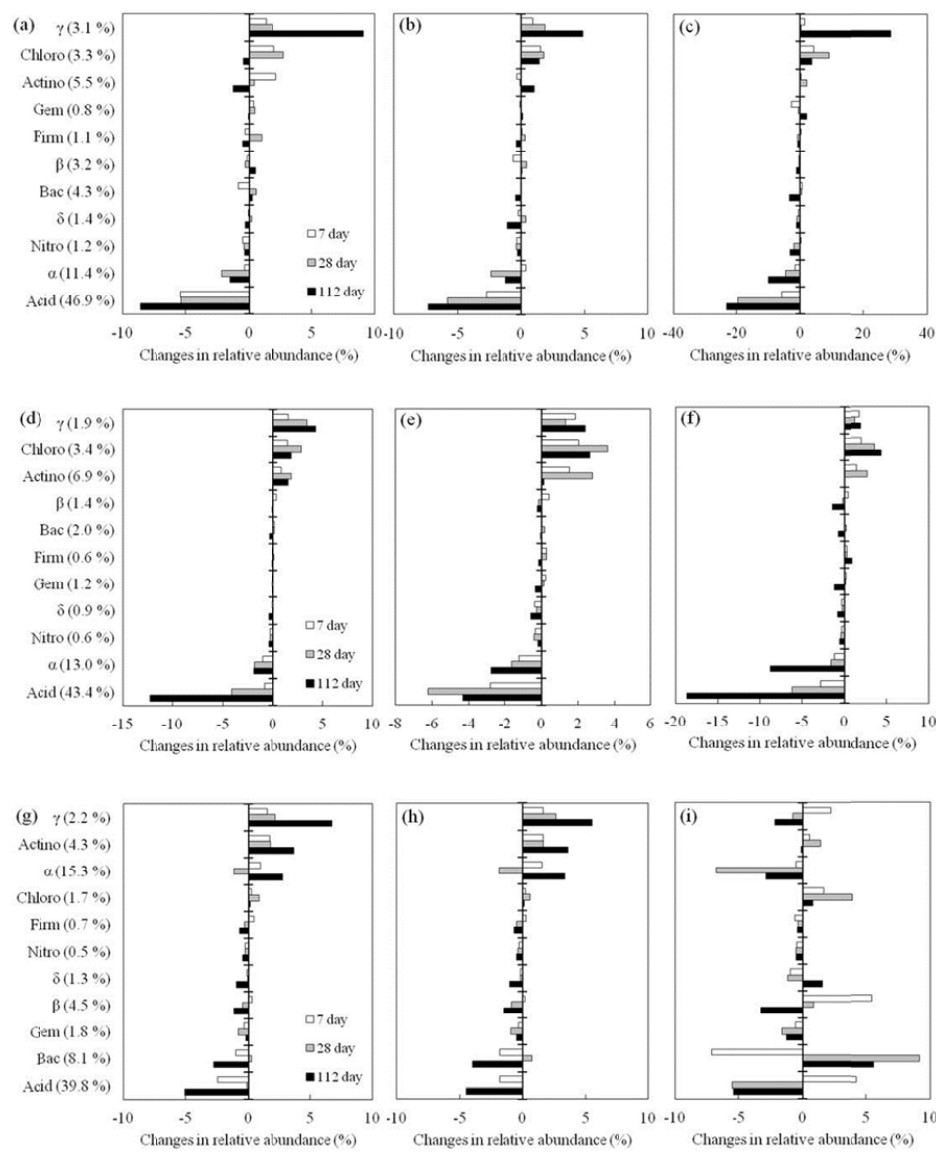



2 **Fig. 6.**

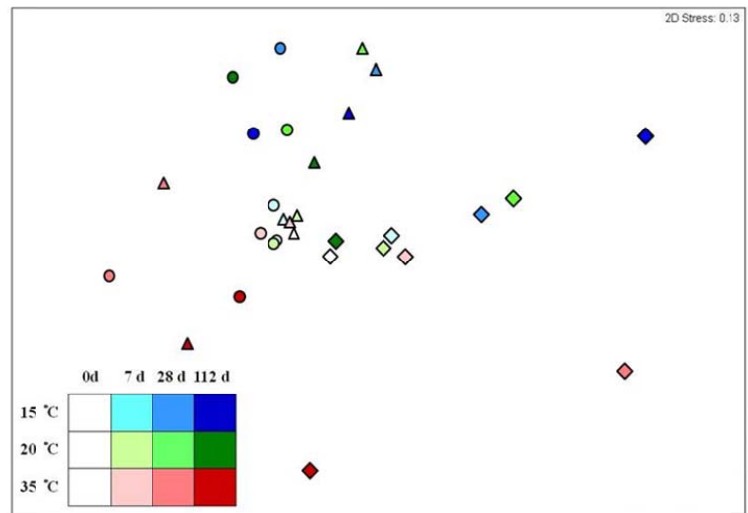





2    **Fig. 7.**

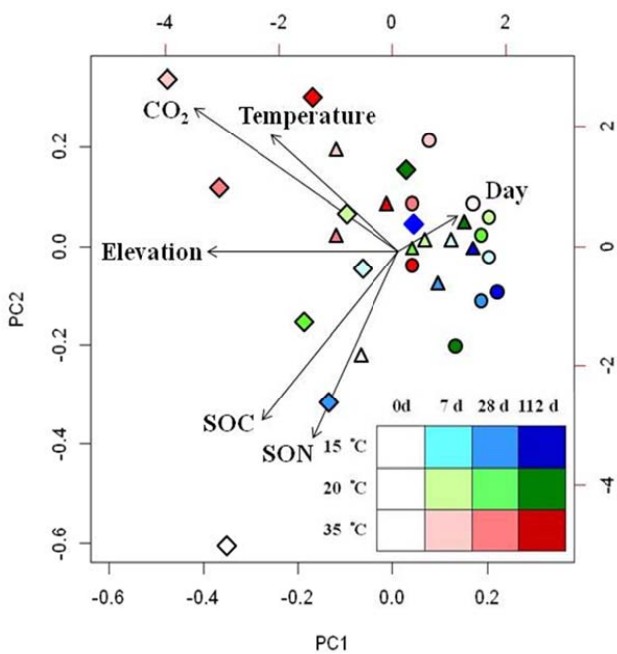

