# Peer review of "Effects of temperature on the composition and diversity of bacterial"

_Biogeosciences, 2017_

## Referee Comment (RC1) · Anonymous Referee #1 · 9 May 2017

**General comments**

Page 2, Line 23: The objectives include determining the response of individual phylogenetic groups to temperature shifts, but there is not data that quantifies phylogenetic groups specifically. The issue with relative abundance is that when abundance of one group increases it could be due to an increase or a decrease in another group. This objective should be revised.

Likewise, the text that discusses Figure 3 and 4 should be qualified to reflect this limitation of abundance data alone. It would greatly strengthen the study if some measure of biomass was taken so that we would at least know how the total population changed during the incubation. Printer-friendly version

Although the introduction and title elude to the fact that the soils being studied were under bamboo, there is no mention of bamboo in the discussion. The title is therefore misleading, as there is no emphasis at all on the plants or on the management of the system. The discussion should be rewritten to include more discussion associated with bamboo and management or the title should be changed.

Specific comments

Page 1, Line 14: I do not know what "a.s.l." is?

Page 1, Line 17-21: The organization of this section needs to be improved. There seems to a sharp juxtaposition from discussing management of bamboo, to discussing abandoned bamboo plots. A strong justification should also be expressed for characterizing microbial communities.

Page 1, Line 24: It isn't clear what a "humpback trend" means in terms of diversity

Page 1, Line 26: It isn't clear what the authors are referring to when they say increase "humification"

Page 2, Line 8-12: There is a lot of challenges in the literature to the copitroph/oligotrophy paradigm. I think there needs to be more literature added to this section. There are also some warming studies that should be referenced here.

Figure 3: There are no error bars on Figure 3, therefore it is difficult to understand what are significant differences.

Page 12, Line 16: It seems like an oversimplification to suggest that the phylum acidobacteria, which has been shown to have a great deal of variation could be used as a climate warming marker. More justification and references should be included to support this statement or it should be removed.

**Technical corrections**

Page 11, Line 16: There appears to be a reference missing for the Oklahoma study.

---

## Referee Comment (RC2) · Anonymous Referee #2 · 12 May 2017

Authors paid attention to the community shift with different incubation treatments using three types of soils collected from 600m, 1200m and 1800m, while measuring the soil respiration, SOC and SON. I think authors well discussed taking the reasonable references supporting the results. However the bacterial groups such as Acidobacteria, Alpha-Proteobacteria and Actinobacteria seemed to be quite broad. So I recommend to discuss more detailed level like order and family. Genus level is quite fine. Then authors can compare the bacterial types those with reference papers. Also your discussion will be more persuasive. Please take in your consideration in revising your paper.

---

## Author Comment (AC1) · 10 Jul 2017

Comment: Authors paid attention to the community shift with different incubation treatments using three types of soils collected from 600m, 1200m and 1800m, while measuring the soil respiration, SOC and SON. I think authors well discussed taking the reasonable references supporting the results. However the bacterial groups such as Acidobacteria, Alpha-Proteobacteria and Actinobacteria seemed to be quite broad. So I recommend to discuss more detailed level like order and family. Genus level is quite fine. Then authors can compare the bacterial types those with reference papers. Also your discussion will be more persuasive. Please take in your consideration in revising

your paper.

Response: We thank the reviewer for the valuable comments. We have added the changes in relative abundance of some abundant genera as shown in Fig. 6. A paragraph has been also added to the Results in p.10, lines 1-12. The discussion of these genera has been elaborated (p. 13, lines 16-20; p. 13, lines 24-26 and p. 14, lines 1-2, and 5-8).

---

## Author Comment (AC2) · 25 Jul 2017

General comments

Page 4, Line 23: The objectives include determining the response of individual phylo-genetic groups to temperature shifts, but there is not data that quantifies phylogenetic groups specifically. The issue with relative abundance is that when abundance of one group increases it could be due to an increase or a decrease in another group. This objective should be revised.

Response: The objective has been revised to "changes in the abundances of different

phylogenetic groups at different incubation temperatures" (p. 4, lines 22-23).

Likewise, the text that discusses Figure 3 and 4 should be qualified to reflect this limitation of abundance data alone. It would greatly strengthen the study if some measure of biomass was taken so that we would at least know how the total population changed during the incubation.

Response: We have clarified that our analyses were based strictly on abundance data in p. 11, lines 10-11, and 16. Unfortunately, we did not have data on biomass during incubation.

Although the introduction and title elude to the fact that the soils being studied were under bamboo, there is no mention of bamboo in the discussion. The title is therefore misleading, as there is no emphasis at all on the plants or on the management of the system. The discussion should be rewritten to include more discussion associated with bamboo and management or the title should be changed.

Response: A paragraph on bamboo and related management practices has been added in the discussion in p. 11, lines 18-22.

Specific comments

Comment: Page 3, Line 14: I do not know what "a.s.l." is?

Response: We have defined the abbreviation, which means: above sea level (p. 3, lines 13-14).

Comment: Page 3, Line 17-21: The organization of this section needs to be improved. There seems to a sharp juxtaposition from discussing management of bamboo, to discussing abandoned bamboo plots. A strong justification should also be expressed for characterizing microbial communities.

Response: This section has been rewritten based on the reviewer's suggestion (p. 3, lines 17-20).

Comment: Page 3, Line 24: It isn't clear what a "humpback trend" means in terms of diversity

Response: It means that the bacterial diversity was less diverse at low and high elevation, with maximum diversity at middle elevations. This is now clearly explained in the text (p. 3, lines 24-25).

Comment: Page 3, Line 26: It isn't clear what the authors are referring to when they say increase "humification"

Response: Our study showed that the bamboo invasion could accelerate the degradation of soil organic matter. We have clarified "invasion of bamboo into adjacent forest soils" in p. 3, line 26 and p. 4, line 1.

Comment: Page 12, Line 8-12: There is a lot of challenges in the literature to the copitroph/oligotrophy paradigm. I think there needs to be more literature added to this section. There are also some warming studies that should be referenced here.

Response: We have elaborated the discussion on copitroph/oligotrophy and warming studies (p. 13, lines 2-3, 5, and 7-8).

Comment: Figure 3: There are no error bars on Figure 3, therefore it is difficult to understand what are significant differences.

Response: Error bars have been added in Figure 3.

Comment: Page 12, Line 16: It seems like an oversimplification to suggest that the phylum acidobacteria, which has been shown to have a great deal of variation could be used as a climate warming marker. More justification and references should be included to support this statement or it should be removed.

Response: This sentence has been deleted.

Technical corrections

[Figure]

Comment: Page 11, Line 16: There appears to be a reference missing for the Oklahoma study.

Response: The missing reference has been added in p. 12, line 6.

Please also note the supplement to this comment:
https://www.biogeosciences-discuss.net/bg-2017-116/bg-2017-116-AC2-supplement.pdf

[Figure]

**Supplement:**

**Response to Reviewers' comments**

**BG-2018-116-RC1**

**General comments**

Page 4, Line 23: The objectives include determining the response of individual phylogenetic groups to temperature shifts, but there is not data that quantifies phylogenetic groups specifically. The issue with relative abundance is that when abundance of one group increases it could be due to an increase or a decrease in another group. This objective should be revised.

**Response:** The objective has been revised to "changes in the abundances of different phylogenetic groups at different incubation temperatures" (p. 4, lines 22-23).

Likewise, the text that discusses Figure 3 and 4 should be qualified to reflect this limitation of abundance data alone. It would greatly strengthen the study if some measure of biomass was taken so that we would at least know how the total population changed during the incubation.

**Response:** We have clarified that our analyses were based strictly on abundance data in p. 11, lines 10-11, and 16. Unfortunately, we did not have data on biomass during incubation.

Although the introduction and title elude to the fact that the soils being studied were under bamboo, there is no mention of bamboo in the discussion. The title is therefore misleading, as there is no emphasis at all on the plants or on the management of the system. The discussion should be rewritten to include more discussion associated with bamboo and management or the title should be changed.

**Response:** A paragraph on bamboo and related management practices has been added in the discussion in p. 11, lines 18-22.

**Specific comments**

**Comment:** Page 3, Line 14: I do not know what "a.s.l." is?

**Response:** We have defined the abbreviation, which means: above sea level (p. 3, lines 13-14).

**Comment:** Page 3, Line 17-21: The organization of this section needs to be improved. There seems to a sharp juxtaposition from discussing management of bamboo, to discussing abandoned bamboo plots. A strong justification should also be expressed for characterizing microbial communities.

**Response:** This section has been rewritten based on the reviewer's suggestion (p. 3, lines 17-20).

**Comment:** Page 3, Line 24: It isn't clear what a "humpback trend" means in terms of diversity

**Response:** It means that the bacterial diversity was less diverse at low and high elevation, with maximum diversity at middle elevations. This is now clearly explained in the text (p. 3, lines 24-25).

**Comment:** Page 3, Line 26: It isn't clear what the authors are referring to when they say increase "humification"

**Response:** Our study showed that the bamboo invasion could accelerate the degradation of soil organic matter. We have clarified "invasion of bamboo into adjacent forest soils" in p. 3, line 26 and p. 4, line 1.

**Comment:** Page 12, Line 8-12: There is a lot of challenges in the literature to the copitroph/oligotrophy paradigm. I think there needs to be more literature added to this section. There are also some warming studies that should be referenced here.

**Response:** We have elaborated the discussion on copitroph/oligotrophy and warming studies (p. 13, lines 2-3, 5, and 7-8).

**Comment:** Figure 3: There are no error bars on Figure 3, therefore it is difficult to understand what are significant differences.

**Response:** Error bars have been added in Figure 3.

**Comment:** Page 12, Line 16: It seems like an oversimplification to suggest that the phylum acidobacteria, which has been shown to have a great deal of variation could be used as a climate warming marker. More justification and references should be included to support this statement or it should be removed.

**Response:** This sentence has been deleted.

**Technical corrections**

**Comment:** Page 11, Line 16: There appears to be a reference missing for the Oklahoma study.

**Response:** The missing reference has been added in p. 12, line 6.

**BG-2017-116-RC2**

**Comment:** Authors paid attention to the community shift with different incubation treatments using three types of soils collected from 600m, 1200m and 1800m, while measuring the soil respiration, SOC and SON. I think authors well discussed taking the reasonable references supporting the results. However the bacterial groups such as Acidobacteria, Alpha-Proteobacteria and Actinobacteria seemed to be quite broad. So I recommend to discuss more detailed level like order and family. Genus level is quite fine. Then authors can compare the bacterial types those with reference papers. Also your discussion will be more persuasive. Please take in your consideration in revising your paper.

**Response:** We thank the reviewer for the valuable comments. We have added the changes in relative abundance of some abundant genera as shown in Fig. 6. A paragraph has been also added to the Results in p.10, lines 1-12. The discussion of these genera has been elaborated (p. 13, lines 16-20; p. 13, lines 24-26 and p. 14, lines 1-2, and 5-8).

---

## Author Response (AR2)

Date of resubmission: September 11, 2017

Editor: Denise Akob
Editor
Biogeosciences

Dear Dr. Akob,

Herewith, I would like to submit a revised version of our manuscript entitled '**Effects of temperature on the composition and diversity of bacterial communities in bamboo soils at different elevations**' (bg-2017-116), to be considered for publication in *Biogeosciences*.

We have thoroughly revised the manuscript according to your suggestions. We would like to gratefully acknowledge the insightful perspectives and suggestions that helped us in improving the manuscript. We hope we have adequately addressed them and that our manuscript is now ready for publication. Please find below our point-by-point responses.

Once again, we would like to thank you for your prompt management and constructive comments on our manuscript, and we are looking forward to hearing from you.

Kind regards,

Chih-Yu Chiu, Ph.D.,
Biodiversity Research Center, Academia Sinica, Taipei, 11529 Taiwan.
Tel: +886-2-2787-1068
E-mail: bochiu@sinica.edu.tw

**Responses to the Editor's comments**

**BG-2018-116-RC1**

**Comments:** Currently, the discussion is highly microbial/soil focused and it would be nice to discuss the implications for plant interactions. Also, link back to management of the former plantations--how will changes in soil respiration/activity affect these legacy sites? or further management of them?

**Response:** We have added some paragraphs in the Discussion to link our findings to plant interactions and management practices:

"In d0 samples, which represent the original composition of the bamboo soils, the bacterial diversity was higher in the 1,800 m soils, followed by the 600 and 1,200 m soils. Communities with higher diversity are reportedly more resistant to environmental changes (Loreau and de Mazancourt 2013). In a study by Ren et al (2015) in rice paddies, the diverse soil communities were more resistant to elevated $CO_2$ and temperature than the less diverse foliar bacterial communities. The increasing concentration of recalcitrant C with increasing elevation (Wang et al., 2016) could be helpful in providing more carbon resources to the community at high elevation. Together, these findings indicate that bamboo soil bacterial communities with higher diversity could be more capable to maintain soil community and function when exposed to climatic changes and subjected to management at high elevation (1,800 m)." (p. 11, line15 to line 26)

"*Bradyrhizobium* of α-*Proteobacteria* at 1,800 m increased at all three incubation temperatures. This genus includes species capable of nitrogen fixation and may significantly contribute to soil function (Yarwood et al., 2009). The increase in their abundance might explain the elevated SON. Moreover, these bacteria are plant growth-promoting bacteria, stimulating plant growth by fixing $N_2$, increasing the availability of nutrients in the rhizosphere, positively influencing root growth and morphology, and promoting other beneficial plant–microbe symbioses (Vessey, 2003). Their response to incubation temperatures indicates their potential roles in bamboo growth and responses to application of fertilizers under climatic change." (p. 14, lines 2–11)

"Within β-*Proteobacteria*, the abundant genus *Burkholderia* is nutritionally versatile and is commonly found in rhizosphere soils. Their functional diversity, including nitrogen fixation and plant growth promotion (Coenye and Vandamme, 2003), could help maintain soil community stability." (p. 14, lines 15–19)

"*Actinobacteria* are involved in the organic matter degradation. Under climatic changes, managements of bamboo forests need to consider the responses of *Actinobacteria* to temperature, especially that in N fertilizers, since the abundance of this phylum was positive affected by N fertilization treatments (Zhou et al., 2015)." (p. 14, lines 22 to line 26)

**Comments:** pg. 4, l. 9: correct spelling to oligotrophs

**Response:** The word has been corrected (p. 4, line 10).

**Comments:** pg. 4, l. 6-9: language is a bit awkward and pluralization is missing.

**Response:** This section has been rephrased as "Soil bacterial communities include different phylotypes that likely represent different functional groups, and their relative abundances are affected by carbon (C) availability. For example, some members of *Proteobacteria* are considered copiotrophs, and their relative abundances appear to be higher in C-rich environments." (p. 4, lines 6–10).

**Comments:** pg. 6: please release the sequences prior to resubmission.

**Response:** The sequences (accession number: SRS1923345) have been released.

**Comments:** pg. 8, l. 4: define day/time notation, e.g., "d28", here to make it clear to the reader for the rest of the paper.

**Response:** We would like to kindly point out that we had defined this notation at first use in the previous revision. To further improve clarity, we have rephrased it as "(after day 28 [d28])" in the current manuscript (p. 8, line 2).

**Comments:** pg. 8, l. 17: OTU's formed is not the correct word choice. Please change to an based on an OTU cutoff of ≤ 0.03

**Response:** We have corrected this accordingly (p. 8, line 15).

**Comments:** pg. 13, l. 15: correct to "life strategies as an oligotroph or copiotroph."

**Response:** We have corrected this accordingly (p. 15, line 11).

**Comments:** Fig. 3: optional--change to a 3 part single column graph

**Response:** Fig. 3 was changed to a 3-part single-column graph.

**Comments:** Fig. 4: please check to see if this is readable for those that are color-blind. As is, the slight variations in darkness of the reds could be difficult to distinguish.

**Response:** In agreement with the editor's apt consideration, we have presented the data originally depicted in Fig. 4 in the newly added Table 1.

[revised manuscript text omitted]